

# TMAC: a Transformer-based partially observable multi-agent communication method

Xuesi Li, Shuai Xue, Ziming He and Haobin Shi

Northwestern Polytechnical University, Xi'an, China

## ABSTRACT

Effective communication plays a crucial role in coordinating the actions of multiple agents. Within the realm of multi-agent reinforcement learning, agents have the ability to share information with one another through communication channels, leading to enhanced learning outcomes and successful goal attainment. Agents are limited by their observations and communication ranges due to increasingly complex location arrangements, making multi-agent collaboration based on communication increasingly difficult. In this article, for multi-agent communication in some partially observable scenarios, we propose a Transformer-based Partially Observable Multi-Agent Communication algorithm (TMAC), which improves agents extracting features and generating output messages. Meanwhile, a self-message fusing module is proposed to obtain features from multiple sources. Therefore, agents can achieve better collaboration through communication. At the same time, we performed experimental verification in the surviving and the StarCraft Multi-Agent Challenge (SMAC) environments where agents had limited local observation and could only communicate with neighboring agents. In two test environments, our method achieves an improvement in performance 6% and 10% over the baseline algorithm, respectively. Our code is available at https://gitee.com/xs-lion/tmac.

## INTRODUCTION

A multi-agent system (*Van der Hoek & Wooldridge, 2008*) is a complex system composed of multiple agents, each of which can perceive, interact, and make autonomous decisions. There are normally three kinds of relationship between agents in systems that are competitive, cooperative, and mixed. Multi-agent systems widely exist in real life and can be used to solve some challenging tasks. For example, multiple robots in a warehouse cooperate to complete cargo sorting (*Dai et al., 2023*); in StarCraft (*Samvelyan et al., 2019*), a large number of different types of unit cooperate to complete tasks such as resource acquisition and attack; in the military field, different equipment and different units conduct coordinated operations; and the latest 6G (*Bazzi & Chafii, 2023*) technology.

Since agents are usually distributed in the environment, partially observable (*Omidshafiei et al., 2017*) has become a common assumption in multi-agent reinforcement learning (MARL). In this case, the agent can only obtain local observations, so it is hard to understand the overall condition of the environment. In addition, MARL is more

Corresponding author
Haobin Shi, shihaobin@nwpu.edu.cn

susceptible to nonstationary (*Papoudakis et al., 2019*) problems because each agent faces a changing environment as well as the changes and adaptation strategies of other agents. However, agents can learn more stably by exchanging information with each other, such as their observations and experiences. Through communication, agents will better understand the environment so that they can better coordinate.

Currently, communication-based multi-agent reinforcement learning (*Zhu, Dastani & Wang, 2022*) methods are usually implemented by adding explicit communication channels between agents based on multi-agent reinforcement learning methods. When establishing a multi-agent communication framework, two core questions must be considered: (1) Who does the agent communicate with? (2) How to generate/utilize messages? Regarding multi-agent communication, a straightforward idea is to communicate between all agents, ensuring that information is shared. However, the importance of collaborative tasks is also different due to differences in status and messages between agents. Full communication may bring a large amount of redundant information, which may interfere with learning collaborative actions while increasing system communication costs (*Hu et al., 2020*; *He et al., 2023*). In multi-agent communication, the messages passed between agents are the key to promoting mutual collaboration. Therefore, the sender of messages should consider how to send more effective messages and the receiver needs to consider how to process messages from other agents to better promote the completion of collaborative tasks.

About how to utilize multi-agent messages in communication, graph convolutional reinforcement learning (DGN) (*Jiang et al., 2018*) abstracts the communication relationship between agents into a graph. The nodes represent the agents, and the edges represent the communication between agents. Meanwhile, DGN uses an attention mechanism to build the convolution layer to extract features. The latent characteristics between agents are used to learn cooperation strategies. Targeted Multi-Agent Communication (TarMAC) (*Das et al., 2019*) and Intrinsic Motivated Multi-Agent Communication (IMMAC) (*Sun et al., 2021*) allow each agent to send signed messages by broadcasting. The agent receiving the message assigns a weight to each message by considering the relevant signatures. TarMAC uses an attention mechanism as the kernel to generate weight, while IMMAC gets weight by softmax. Graph attention (GA) Comm (*Liu et al., 2020*) learns a soft and a hard attention layer, one for choosing agents to communicate with and the other for determining the relative importance between agents. GA Comm also learns a graph neural network (GNN) to aggregate messages. BiCNet (*Peng et al., 2017*) proposes a method that connects each agent's policy and value functions through a bidirectional long short-term memory (BiLSTM) layer. Consequently, agents can capture varied memory states that exhibit long-term dependencies and adapt their message exchanges accordingly. The MD-MADDPG framework (*Pesce & Montana, 2020*) facilitates the ability of an agent to retain a shared memory that serves as the context of its environment. Subsequently, the agent acquires skills to sequentially read from and write to this memory, akin to operations within a long short-term memory (LSTM) model. While existing research predominantly concentrates on the receiver's management of messages

from other agents during communication processes, it largely overlooks strategies by which message senders can enhance the effectiveness of their communications.

In order to solve the problem above, in this article, we propose the Transformer-based (*Vaswani et al., 2017*) multi-agent communication (TMAC), which uses a Transformer to learn message features from neighboring agents and further helps the agent process its own observations and message features from neighboring agents. In summary, our main contributions are as follows.

1) We propose a novel multi-agent collaboration algorithm called TMAC, which can efficiently extract information from agents. It uses a transformer as a core component to help agents extract features from messages of other agents and generate their output messages.

2) We propose a self-message fusion algorithm for each agent. The transformer helps the agent fuse the messages, thereby sending more efficient messages in the communication, enabling the agent to obtain multiple forms and dimensions of observations in the environment.

3) We evaluate our algorithm in a multi-agent grid environment called Surviving, which requires agents to cooperate to achieve higher scores. Results of the experiment show that our algorithm has better performance in environments of different difficulties compared to baselines.

## RELATED WORK

Coordinating agents in multi-agent systems poses a significant challenge. Within a multi-agent system, agents observe the environment and each other, rendering the environment dynamic for every agent involved. Moreover, the complexity of coordinating the overall system is further compounded by the challenge of distributing reputation among agents, making it hard for each agent to assess its impact on the system. In the subsequent sections, we will delve into recent advancements in multi-agent reinforcement learning to address these challenges.

### Communication-based multi-agent reinforcement learning

In the Centralized Training Decentralized Execution (CTDE) framework, although some problems of unstable environments can be overcome through centralized training because during the execution process, each agent can only obtain its local observations, there are still difficulties in collaboration between agents. To address this problem, many scholars have adopted communication-based multi-agent reinforcement learning methods to introduce information transfer between agents. This part of the research mainly focuses on cooperative multi-agent systems. Intuitively, when there is a need for collaboration between agents, and each agent can only obtain its local observations, information transfer between agents will be conducive to completing the collaborative task. On the other hand, through information transfer between agents, local centralization can be achieved, thereby further overcoming the problem of non-stationary environments and promoting collaboration between agents.

Works such as Differentiable Inter-Agent Learning (DIAL) (*Foerster et al., 2016*) and Reinforced Inter-Agent Learning (RIAL) (*Foerster et al., 2016*) allow communication between multiple agents in cooperative scenarios. Work in RIAL and DIAL is appropriate for communication when resources are limited, enabling agents to exchange binary or real-valued messages. CommNet extends DIAL to a continuous communication protocol tailored for tasks that require full collaboration. In this protocol, agents receive encoded hidden states from other agents and utilize the messages to inform their decision-making process. Nevertheless, aggregating the messages through summation or averaging could lead to information loss.

Some early studies suggest agents use a gate mechanism to decide whether to exchange information with other agents. Attentional Communication Model (ATOC) (*Jiang & Lu, 2018*) proposes that agents only communicate with some agents within the observable range. A probability gate determines which surrounding agents can be assigned to a group for communication. After CommNet, IC3Net (*Singh, Jain & Sukhbaatar, 2019*) also uses a gate mechanism to decide whether agents should send a message to other agents. In addition, IC3Net adopts personalized rewards for each agent instead of the global shared rewards, thereby exhibiting more diverse behaviors in competitive environments. I2C measures the contingent effects of considering actions and strategies from other agents. Each agent will then decide whether to communicate with others peer-to-peer.

Compared to the flexible decisions provided by gating units for communication, we can consider explicitly constructing a global communication graph to optimize the information exchange between agents. SchedNet (*Kim et al., 2019*) suggest selecting some of the agents by learning to broadcast their messages. Multi-Agent Graph-attentIon Communication (MAGIC) (*Niu, Paleja & Gombolay, 2021*) and FlowComm (*Du et al., 2021*) focus on facilitating agent communication by developing a shared graph. This graph assists agents in making informed decisions regarding the necessity of communication and choosing appropriate communicating parties. GA-Comm employs an attention mechanism to determine which pairs of agents can communicate with each other. On the other hand, MAGIC and Flow-Comm achieve more refined control by constructing directed graphs among agents. This structure enables connected agents to engage in unilateral or bilateral communication with others.

Connectivity Driven Communication (CDC) (*Pesce & Montana, 2023*) dynamically alters the communication graph through a diffusion process perspective, and temporal cooperative weighted graph convolution MARL method (TWG-Q) (*Liu et al., 2022*) emphasizes temporal weight learning and the application of weighted graph convolution network (GCN), CommFormer adopts a different approach. It focuses on learning a static graph, aimed at optimizing communication efficiency prior to the inference phase, setting it apart from the traditional methodologies employed by the aforementioned approaches.

## Attention mechanism in multi-agent reinforcement learning communication

The attention mechanism is similar to the way the human brain processes messages. After receiving many messages, the attention mechanism can assign corresponding weights to

the messages, so more resources can be invested in important parts in the subsequent processing. In early research on multi-agent reinforcement learning communication, such as CommNet, agents processed the messages they received equally, and these messages were averaged and used as the basis for the agent's decision-making. In subsequent work, the attention mechanism has been widely used.

Multi-Actor-Attention-Critic (MAAC) (*Iqbal & Sha, 2019*) is based on the actor-critic algorithm. During training, each agent uses the attention mechanism to ask other agents for the other's observation information, obtains the corresponding weight coefficient, and weights and sums this information as the input of its own critic to help The agent make decisions. G2ANet (*Liu et al., 2020*) uses a hard attention mechanism to dynamically learn the interactive relationship between agents and cut off the communication between some agents, thereby reducing the communication overhead. The soft attention mechanism employed simultaneously assigns weights to input information in order to enable the agent to focus more on messages that are highly relevant to itself. In the DGN (*Jiang et al., 2018*) model, the multi-agent environment is depicted as a graph, with the attention mechanism serving as a convolution kernel for extracting relationship features among agents. And other Excellent works like, Actor-coordinator-critic Net (ACCNet) (*Mao et al., 2017*), the omission of early-era deep multi-agent (MA) communication, VBC (*Zhang, Zhang & Lin, 2019*) MA communication with limited-bandwidth and PoSD (*He et al., 2024*).

# PREPARATION WORK

## Partially observable Markov games

Markov game and decentralized partially observable Markov decision process (Dec-POMDP) (*Littman, 1994*) are two models commonly used for modeling multi-agent systems. Dec-POMDP is designed explicitly for partially observable environments. For instance, in the case of a fully cooperative multi-agent system, Dec-POMDP can be utilized to represent the system in the following manner:

- N: In a multi-agent system, the number of agents is typically equal to or greater than 2. This ensures that multiple entities interact within the system;
- S: State space contains information about the agent and the environment;
- $A = [A_1, A_2, \ldots, A_N]$: Joint action space, which $A_i$ represents the set of local actions that an agent can take $a_i$;
- $T(s'|s, a) : S \times A \times S \rightarrow [0, 1]$: State transition function describes the probability of transitioning from one state to another after the agents take a joint action in the current state. It provides insights into the dynamics of the system.;
- $R : S \times A \times S \rightarrow \mathbb{R}$: Reward function is used to describe the reward obtained by the agent when a state transition occurs. In a fully cooperative environment, all agents share the same reward function;
- $O = [O_1, \ldots, O_N]$: set of joint observations;
- $Z : S \times A \rightarrow O$: observation function;
- $\gamma \in [0, 1]$: Discount factor.

In the context of Dec-POMDP, the agent is restricted to acquiring local observations and making decisions solely based on these local observations (or information exchanged with other agents) within a specific state. This process leads to a transition of the environment to a different state, followed by receiving a collective reward. The agent's primary objective is to develop a strategy that optimizes the discounted return.

## DQN

Q-learning, an algorithm for reinforcement learning without a model, was first introduced by Watkins in 1989. Q-learning has been proven to determine the optimal policy for any finite Markov decision process (MDP). The primary objective of Q-learning is to maximize the expected value of the total reward obtained from all subsequent steps, starting from the initial state. During the initialization phase, the Q-values are set to a predefined constant. Subsequently, at each time step $t$, an agent $i$ chooses an action $a_i^i$, receives a reward $R_t$, transitions to a new state $S_{t+1}$, and updates the Q-value accordingly. The fundamental operation of this algorithm is encapsulated in its core Eq. (1), which governs these dynamics.

$$Q(s_t, a_t) \leftarrow Q(s_t, a_t) + \alpha \cdot [r_t + \gamma \max_\pi Q(s_{t+1}, a_t) - Q(s_t, a_t)], \tag{1}$$

where $\alpha$ is the learning rate. The DQN (*Van Hasselt, Guez & Silver, 2016*) algorithm is a technique that utilizes a neural network to approximate the value function of Q-Learning. In this algorithm, the value function is denoted as $Q(s, a, \theta)$, where $\theta$ represents the parameters within the neural network. Consequently, the primary focus of the learning process lies in determining the optimal values for $\theta$ to effectively approximate the value function. In the case of the gradient descent method, the loss function is defined by Eq. (2).

$$L_i(\theta_i) = E_{(s,a,r,s^i) \sim U(D)}[r + \gamma \max Q(s', a^i; \theta_i^-) - Q(s, a; \theta_i))^2]. \tag{2}$$

Among them, $\theta_i^-$ is the target network parameter of the i-th iteration, $\theta_i$ is the network parameter, and $D$ represents the experience replay pool stored during the training process.

## METHOD

This section introduces the multi-agent communication collaboration algorithm and the agent message self-fusion module proposed in this article. We assume that the current multi-agent environment is partially observable. There are $N$ agents in the environment. At each time $t$, the agents can only observe the environment within a certain distance around them. Moreover, message passing between agents is limited to a certain range.

### Overview

The multi-agent reinforcement learning communication process framework we proposed is shown in Fig. 1. At time $t$, each agent $i$ gets its own local observation $o_i^t$ from the environment. At the same time, we can obtain the communication adjacency matrix $G^t$ at the current moment. This matrix defines a group of agents that can complete an

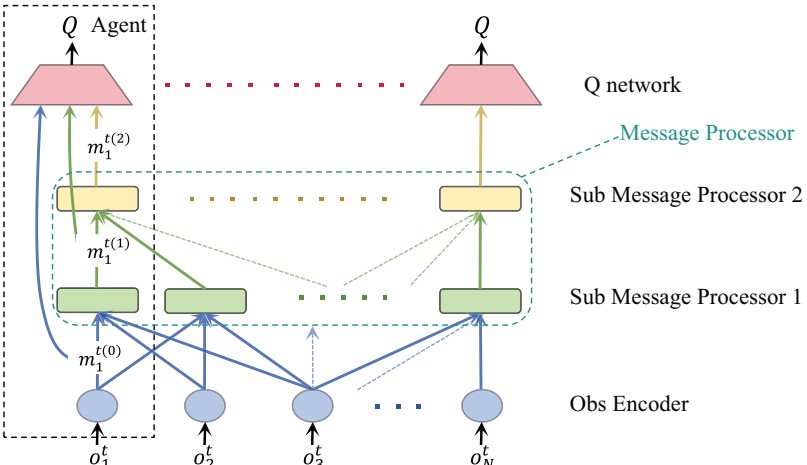

**Figure 1** Flow chart of our proposed multi-agent reinforcement learning communication framework.

information exchange among all agents at the current moment. The original observation of the agent is first initially encoded by Obs Encoder, $m_i^{t(l)}$ as shown below:

$$m_i^{t(0)} = f_{OE}(o_i^t), \tag{3}$$

$m_i^{t(0)}$ represents the message of the *i-th* agent. We call the stage in which the current message is not processed as round 0, so the symbol is given $t(0)$.

As shown in Fig. 1, we define a message processor module that helps the agent process messages from other agents. The message processor receives the output from the observation coding layer $m_i^{t(0)}$ and the agent communication adjacency matrix in the current environment $G^t$ as input and outputs the processed and fused message $m_i^{t(l)}$. A message processor is composed of multiple sub-message processors. By stacking the number of sub-message processors, the receptive field of the agent will continue to expand, thereby collecting more information and collaborating with a wider range of agents. Through a sub-message processor, agent *i* can obtain their observation input from agents within one hop. By stacking two sub-message processors, agent *i* can obtain the output of the first sub-message processor of the agent within one hop, which contains the information of the two-hop agents. However, no matter how many sub-message processors are stacked, the agent can only communicate with adjacent agents each time, which makes this method more suitable in practice.

After multiple rounds of information exchange, we concat the outputs of all previous layers and input them into Q Net for each agent so that the intelligent agent can accept observations and characteristics from different fields and make more rational decisions.

In summary, during a single TMAC processing cycle involving multiple agents, messages from various sources are first passed through the self-message fusion module of each agent. This module standardizes the messages into a unified dimension and performs an initial concatenation. Subsequently, the messages are encoded using an encoder to produce processed representations. These encoded messages are then passed to the

message processor, which comprises multiple sub-processors. Each sub-processor receives all message inputs from the previous layer and uses a multi-head attention mechanism to weight and fuse the inputs, resulting in processed outputs. Finally, these refined messages are fed into each agent's Q-network to estimate action values.

During the training process, we save tuples $\{O, A, O', R, C\}$ into the replay buffer for each time step. Here, $O' = \{o'_1, \ldots, o'_N\}$ represents the set of agent observations at the next moment, and $C = \{C_1, \ldots, C_N\}$ denotes the set of adjacency matrices between agents. Subsequently, we randomly choose a minimum batch S from the replay buffer and aim to minimize the loss as defined in Eq. (4).

$$\mathcal{L}(\theta) = \frac{1}{S}\sum_{S}\frac{1}{N}\sum_{i=l}^{N}\left(y_i - Q(o_{i,\mathscr{C}}, a_i; \theta)\right)^2. \tag{4}$$

## Message processor

Message processors help agents aggregate received messages to make more beneficial decisions. The message processor receives the encoded message $\{m_i^{t(0)}\}_1^N$ and the adjacency matrix $G^t$ between the agents and outputs the processed message. The process is described in Eq. (5). As introduced before, a message processor is generated by stacking multiple sub-message processors. The flow of messages in a sub-message processor is equivalent to a round of communication. The results of a sub-message processor are shown in Fig. 2.

$$\{m_i^{t(L)}\}_1^N = f_{MP}(m_1^{t(0)}, \cdots, m_N^{t(0)}, G^t). \tag{5}$$

The sub-message processor consists of a multi-head attention module, a multi-layer perception module, two residual connections, and two normalization layers. $M^{t(l)} = \{m_0^{t(l)}, \ldots, m_N^{t(l)}\}$ represents the message passed to the *l-th* layer sub-message processor, which is composed of message vectors of multiple agents. $G^t$ represents the adjacency matrix of multiple agents in the environment at time t, which determines the amount of information the agents can exchange in a round of communication.

In the multi-head attention module, the input feature vector of the agent is first projected into a query vector (query), key (key), and value representation (value) through each attention head. Let $\mathbb{B}_{+i}$ represent $\mathbb{B}_i$ and $i$. For attention head m, the relationship between $i$ and $j$ belonging to $\mathbb{B}_{+i}$ (attention weight) can be calculated as:

$$\alpha_{ij}^m = \frac{\exp(\tau \cdot \mathbf{W}_Q^m h_i \cdot (\mathbf{W}_K^m h_j)^\top)}{\sum_{k\in\mathbb{B}_{+i}}\exp(\tau \cdot \mathbf{W}_Q^m h_i \cdot (\mathbf{W}_K^m h_k)^\mathrm{T})}. \tag{6}$$

The Eq. (6) is used to correlate the query vector of agent *i* with the keys of all neighbor nodes that can communicate and then perform the softmax function to obtain the attention weight. Among them, $W_m^Q$, $W_m^K$, and $W_m^V$ respectively represent the parameter matrix corresponding to the query vector, key, and value representation in the attention head m, which is the scaling factor and is usually set to the reciprocal root of the feature

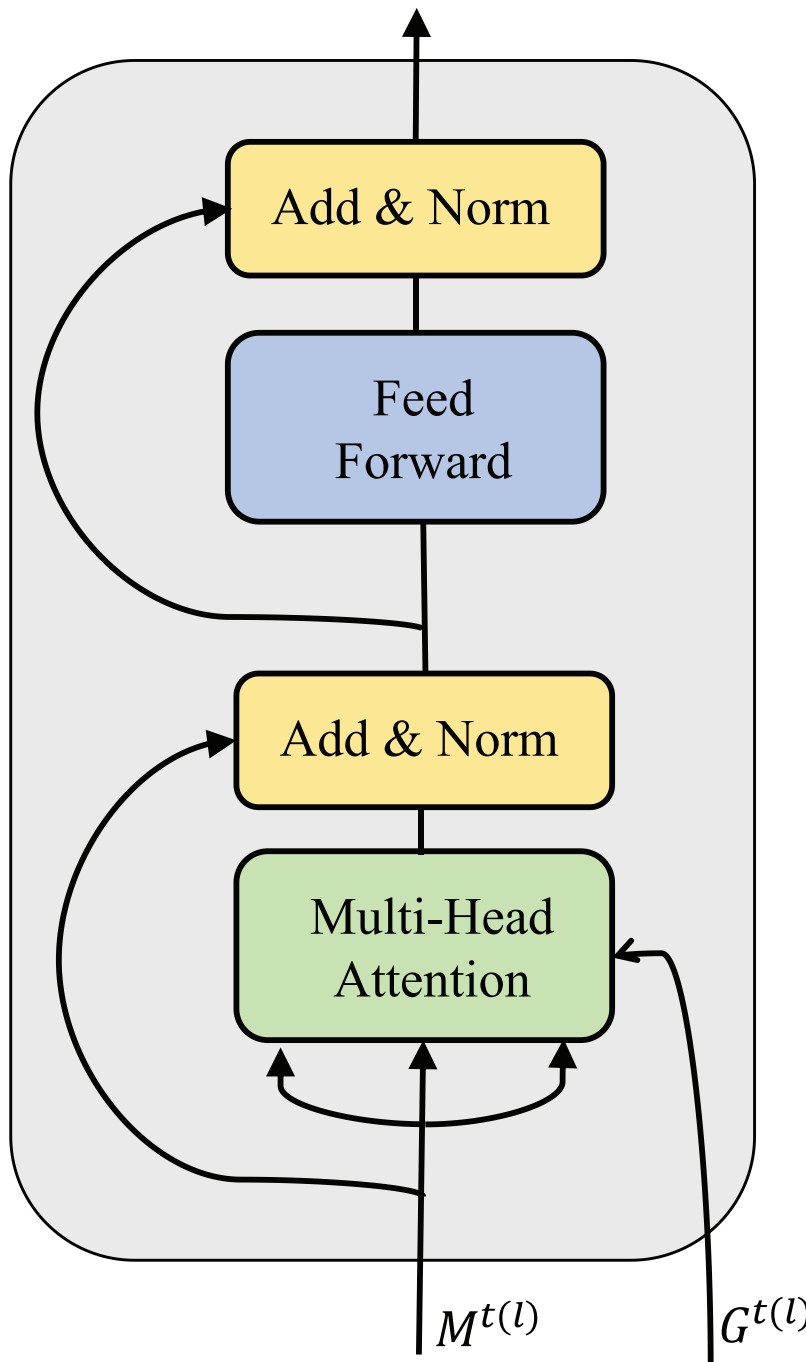

**Figure 2** Structural diagram of sub-message processor.

vector dimension. Then, the input and output residuals of agent *i* in this module are connected and normalized to retain the original features and gradients. The normalized data is input into the multi-layer perception. The hidden layer is used to further extract the feature correlation in the agent message, and the input and output are residualized and normalized to obtain the final output.

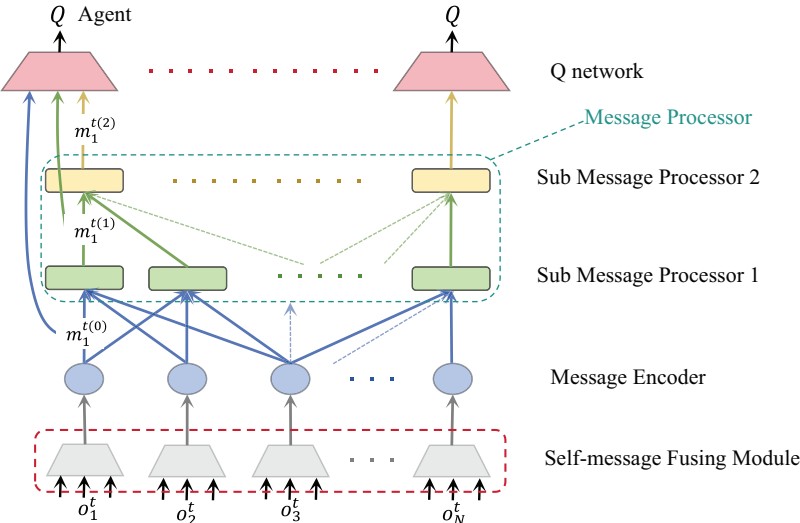

**Figure 3  Communication flow chart after adding the self-message fusion module of agent.**

## Agent self-message fusion module

In multi-agent communication, in addition to the receiver needing to consider how to process messages from other agents, the senders of the messages need to consider how to send more effective messages to promote the completion of collaborative tasks better. The self-message fusion module of the agent helps the agent extract the characteristics of its own observation. It helps the agent determine what content should choose to participate in communication. Considering the self-message fusing module as a black box, it takes observation $\{o_i^{t(0)}\}_1^K$ from the agent and outputs the processed $e_i^t$, we represent the self-message fusing module in Eq. (7).

$$e_i^t = f_{SMF}(o_i^{t(0)}, \ldots, o_i^{t(K)}). \tag{7}$$

In overview subsection, we directly input the observations of agents from the environment into the encoding layer and further fuse the messages. Considering that in a complex environment, the agent receives more observations, which may come from different observations, such as radar, camera information and others. It is difficult for a simple multi-layer perception to extract enough features, thereby affecting the agent carrying out subsequent communication and action selection. Therefore, in response to this situation, we propose to process the observation of the agent and extract message features from multiple sources.

After applying the sub-message fusion module to our algorithm, the communication framework is shown in Fig. 3. It shows that each agent obtains multiple dimensions of information from the environment, which can be original environmental data or artificially divided observation data. These original environmental data contain the observation of agents. Before communicating with other agents, we hope the agent can extract key information from these data. Therefore, this part is to help the agent select and

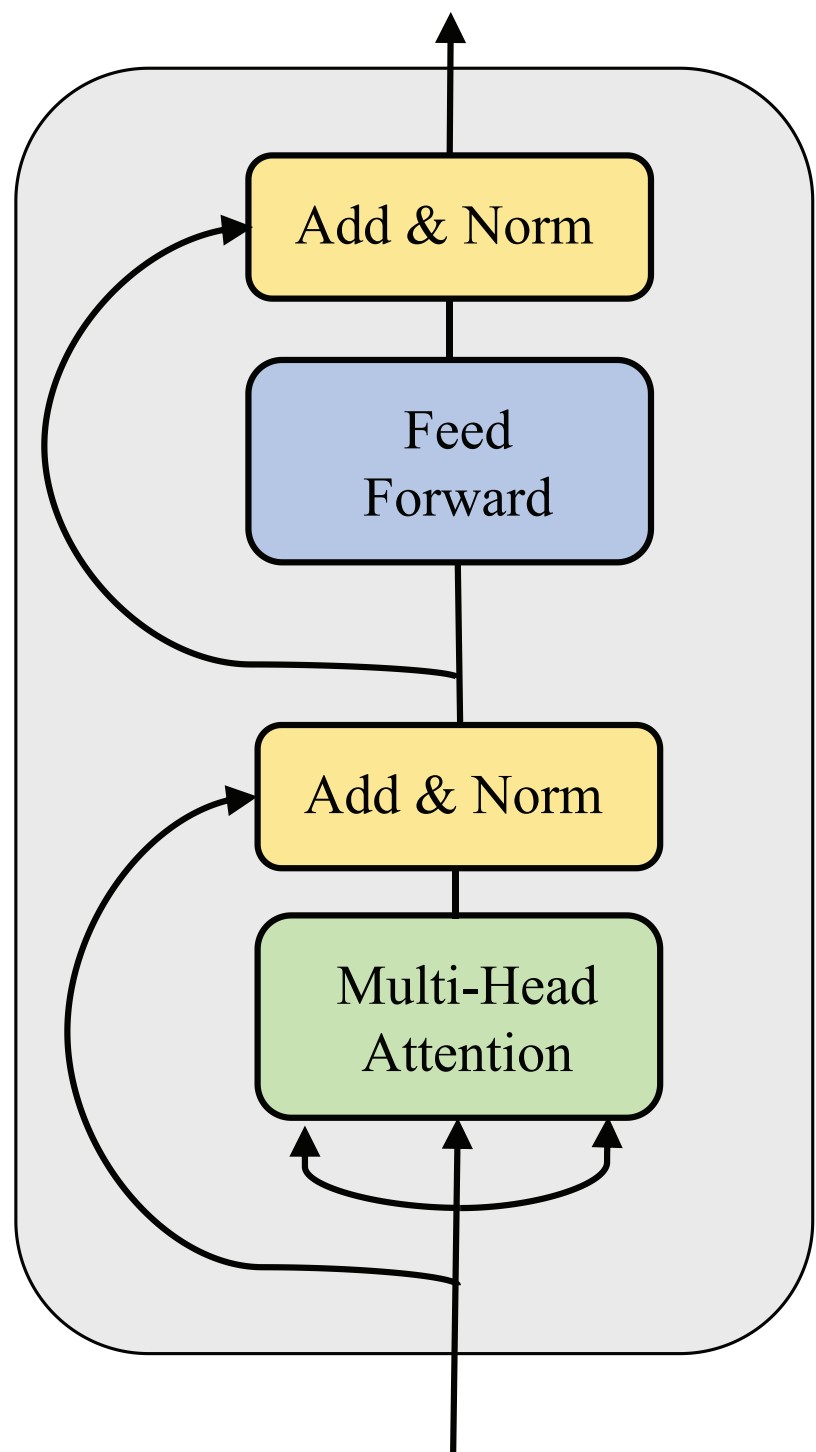

**Figure 4** Diagram of network structure of the agent self-message fusing module (without adjacency matrix $G^t$ compared to **Fig. 2**).

integrate the content that should be used for communication. More efficient message equation will help improve the communication efficiency of the overall system to a certain extent.

---

**Algorithm 1** Training procedure for TMAC.

1 Initialize max updates $M$, agents $N$, max steps in an episode $T$, update epochs after an episode $E$;

2 Initialize value function $Q$ with random weights $\theta$

3 Initialize target value function $\hat{Q}$ with weights $\theta^- = \theta$

4 **for** *episode* $\leftarrow 1$ **to** $M$ **do**

5     Initialize environment, get observation $\{o_i^{t(k)}\}_1^K$ for each agent $i$ and communication adjacency matrix $G^t$. For better description, we use $O_i^t$ as $\{o_i^{t(k)}\}_1^K$;

6     **for** $t \leftarrow 1$ **to** $T$ **do**

7         Get observation after self-message fusing module $e_i^t = f_{SMF}(o_i^{t(0)}, \ldots, o_i^{t(K)})$ for each agent $i$;

8         Get message at communication round(0) $m_i^{t(0)} = e_m(e_i^t)$ for each agent $i$;

9         Get processed message $\{m_i^{t(l)}\}_1^N = f_{MP}(m_1^{t(0)}, \ldots, m_N^{t(0)}, G^t)$;

10         Calculate q-value vector $\{q_i^t\}_1^N = MLP(m_i^{t(l)})$ for each agent $i$;

11         With probability $\varepsilon$ select a random action $a_i^t$, otherwise select $a_i^t = argmax_a(q_i^t)$;

12         Execute action $a_i^t$ in emulator and observe reward $r_i^t$ and $\{O_i^{t+1}\}_1^N$;

13         Store transition $(\{O_i^t\}_1^N, a_i^t, r_i^t, \{O_i^{t+1}\}_1^N, terminate)$ in Replay Buffer;

14     **end**

15     **if** *episode* $\geq 100$ **then**

16         **for** $e \leftarrow 1$ **to** $E$ **do**

17             Sample random mini-batch of transitions $(\{o_i^{t(k)}\}_1^K, a_i^t, r_i^t, \{o_i^{t+1(k)}\}_1^K, terminate)$ from Replay Buffer;

18             Perform a gradient descent step on with respect to the network parameters $\theta$;

19             Set $y_j = \begin{cases} R_j & \text{if episode terminates} \\ R_j + \gamma \max_{a'} \hat{Q}(\{O_i^{t+1}\}_1^N; \theta^-) & \text{otherwise}; \end{cases}$

20             Perform a gradient descent step on $(y_j - Q(\{O_i^t\}_1^N; \theta))^2$ with respect to the network parameters $\theta$;

21         **end**

22     **end**

23     Every $C$ steps reset $\hat{Q} = Q$;

24 **end**

---

For each agent, their observations are encoded into multiple vectors, which are fed into the message fusion module and finally output a selected and integrated message equation. The network structure of the self-message fusion module is shown in Fig. 4. Meanwhile, the training procedure for our algorithm is shown in Algorithm 1.

# EXPERIMENTS AND RESULTS
## Simulation environment
### Surviving

For the experimental part, the simulation environment we used comes from the surviving multi-agent grid simulation environment in the DGN framework. There are multiple agents in this environment that interact with the environment; Each agent corresponds to a grid and has a different limited local observation that contains a square view of a $n \times n$ grid centered on the agent. This agent can communicate with neighboring agents in a

**Table 1 Map size, communication distance, and number of agents settings at different difficulty levels in the Surviving simulation environment.**

| Difficulty | Map size | Communication distance | Number of agents |
|---|---|---|---|
| Easy | $14 \times 14$ | 3 | 10 |
| Medium | $18 \times 18$ | 4 | 20 |
| Hard | $24 \times 24$ | 5 | 30 |

square area with a grid $m \times m (m < n)$. At each time step, each agent can move to one of the four adjacent grids or eat food at its location. Each agent has its own health value $HP_0^t = 10$, and their rewards are also assessed based on their health value. The specific details are as follows.

$$HP_i^t = \begin{cases} HP_i^t - 1, & action = move \\ HP_i^{t-1} + food[x,y], & action = eat \end{cases}$$

$$r_i^t = \begin{cases} -0.2, & HP_i^t \leq 0 \\ 0.4, & HP_i^t > 0 \end{cases}.$$

The difficulty of the task increases with the number of grids and agents. We have set three different difficulties for this environment. The specific environment configuration is shown in Table 1.

### SMAC

At the same time, we also tested our algorithm on the StarCraft Multi-Agent Challenge (SMAC) (*Samvelyan et al., 2019*) environment. SMAC is WhiRL's environment for research in the field of cooperative multi-agent reinforcement learning (MARL) based on Blizzard's StarCraft II RTS game. SMAC makes use of Blizzard's StarCraft II Machine Learning API and DeepMind's PySC2 to provide a convenient interface for autonomous agents to interact with StarCraft II, getting observations and performing actions. Unlike the PySC2, SMAC concentrates on decentralized micromanagement scenarios, where each unit of the game is controlled by an individual RL agent.

### Baselines

We selected DGN (*Jiang et al., 2018*), and MAGIC (*Niu, Paleja & Gombolay, 2021*) as baseline algorithms to compare with our algorithm. DGN is a traditional communication algorithm for multi-agent reinforcement learning. It suggests conceptualizing multi-agent communication as a graph, with each agent represented as a node. The attributes of the node depict the agent's local observation. Nodes are connected when there exists an edge between adjacent nodes. DGN uses the multi-head attention mechanism as the convolution kernel to extract potential features between agents to learn cooperation strategies. MAGIC proposes (1) using a scheduler to solve the problem of who the agent communicates with and when and (2) using message processors and dynamic graphs to process communication signals. Make the entire system more efficient through fine-grained communication control.

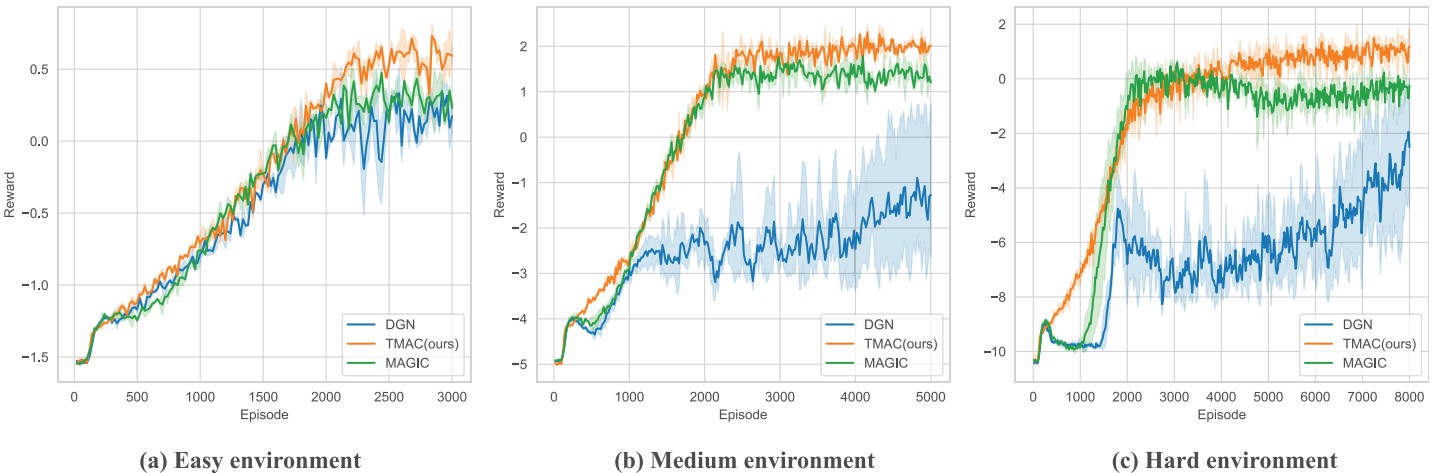

(a) Easy environment      (b) Medium environment      (c) Hard environment

**Figure 5** Performance comparison chart of our algorithm and baseline (DGN, MAGIC) under easy, medium and hard environment settings.

In our algorithm, the agent encodes its own original observations and passes them through the message self-encoding module, extracting valid messages for subsequent communication. The message extracted by each agent is sent to the message processor. The message processor is composed of multiple sub-message processors. The sub-message processors help the agent process and aggregate the received messages. The messages passing through the message processor are sent to QNet to obtain the corresponding Q value output. This helps the agent choose more beneficial actions.

Considering the settings in DGN and MAGIC, we configured one Transformer encoder as a sub-processor, with a total of two encoders. For the specific number of layers within each Transformer encoder, we chose two layers. This choice was made to reduce computational resource requirements while ensuring a fair comparison in terms of the total number of parameters.

## Experimental results

### Performance verification

As we can find in Fig. 5 and Table 2, our algorithm performs better than the baseline under different environment difficulty settings. Among them, the performance gap between our algorithm and the baseline algorithm is small in a simple environment. This is because the map size in a simple environment is small, and the agent can obtain a lot of information through observations to support its strategy selection. As the difficulty of the environment increases, our algorithm performs better, outperforming the best baseline algorithm by 5% and 9% in medium and hard environments, respectively.

This is because the communication of agents is more important in a large map environment. Our algorithm can better help the agents judge what messages they should send and assist the agents in processing the received messages so that the entire system can collaborate better. Our algorithm performs better than the baseline under different environment difficulty settings. Among them, the performance gap between our algorithm

**Table 2 Table of average rewards obtained by our algorithm *vs.* the baseline algorithm in surviving and SMAC.**

|        | DGN          | MAGIC        | TMAC (Ours)      |
|--------|--------------|--------------|------------------|
| Easy   | −0.44 ± 0.03 | −0.36 ± 0.01 | **−0.25 ± 0.01** |
| Medium | −2.66 ± 0.14 | −0.10 ± 0.05 | **0.22 ± 0.02**  |
| Hard   | −6.97 ± 0.66 | −2.19 ± 0.02 | **−1.33 ± 0.3**  |
| 25 m   | 7.22 ± 0.47  | 4.21 ± 0.08  | **13.97 ± 0.32** |
| MMM2   | 7.35 ± 0.25  | 4.85 ± 0.85  | **8.74 ± 0.2**   |

**Note:**
Bold indicates the best results.

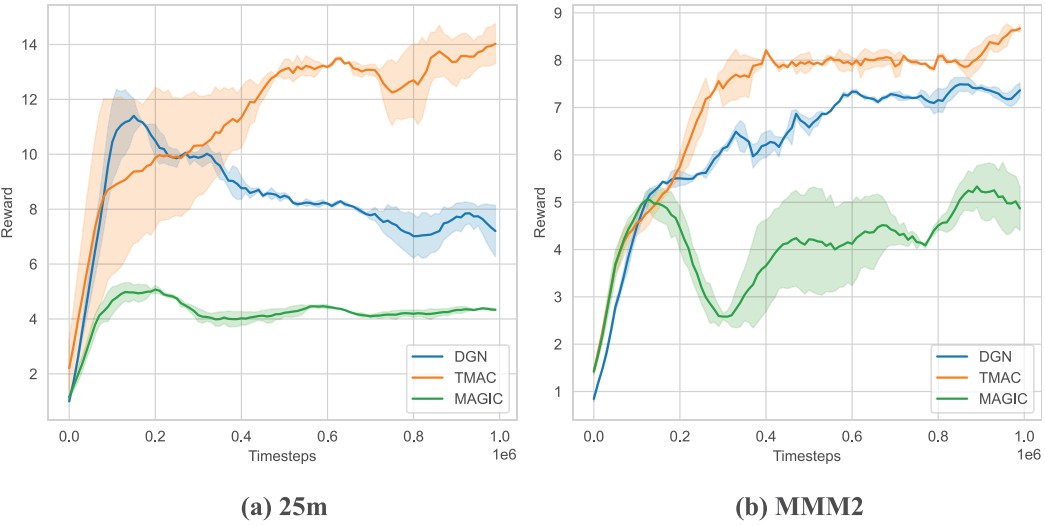

(a) 25m                                      (b) MMM2

**Figure 6 Performance comparison chart of our algorithm on SMAC.**

and the baseline algorithm is small in a simple environment. This is because the map size in a simple environment is small, and the agent can obtain a lot of information through its observations to support its strategy selection.

As shown in Fig. 6 and Table 2, we tested our algorithm in two maps of SMAC: 25 m and MMM2. The results show that our algorithm outperforms the baseline algorithm on two maps and can obtain higher rewards. At the same time, we found that as the difficulty of the map increases (MMM2 is more difficult than 25 m), the improvement of our algorithm will be smaller. This is because our algorithm uses fewer parameters and does not have sufficient representation capabilities for more complex environments. At the same time, as the difficulty increases, the agent needs to train for longer rounds and explore more, but here we use fewer steps only to show the performance of the algorithm.

### Ablation study results

After verifying the overall performance of our algorithm, we perform ablation testing of the modules within it. As shown in Figs. 7, 8 and Table 3, we removed the self-encoding module and the message processor module, respectively. It should be noted that when
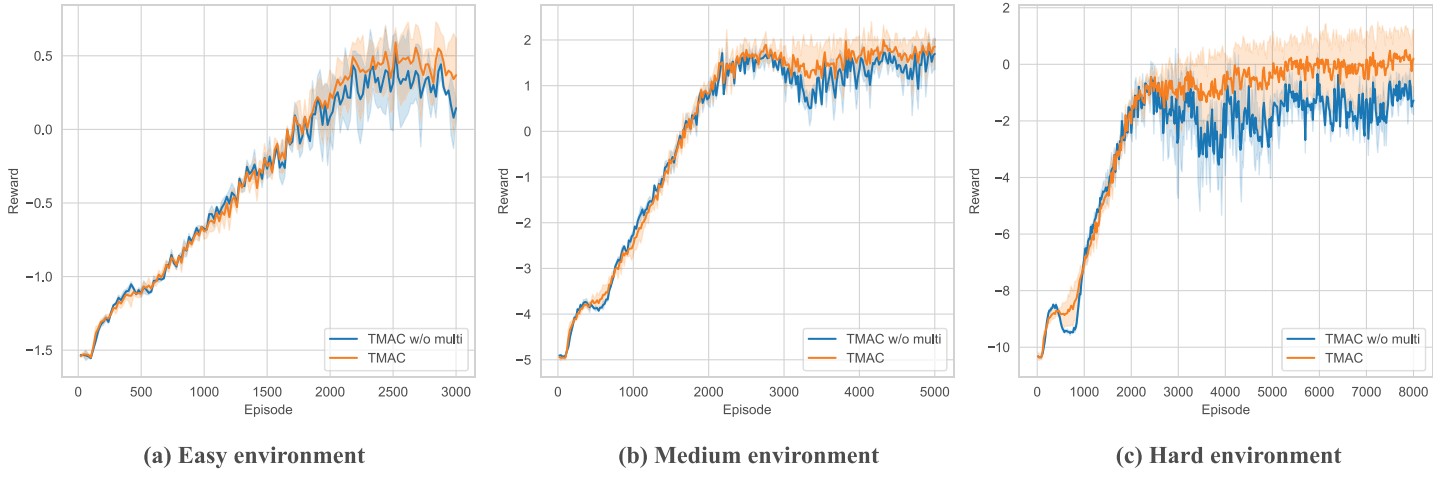

**(a) Easy environment**  **(b) Medium environment**  **(c) Hard environment**

**Figure 7** Performance comparison diagram of our algorithm after removing the self-message fusing module in easy, medium and hard environments.

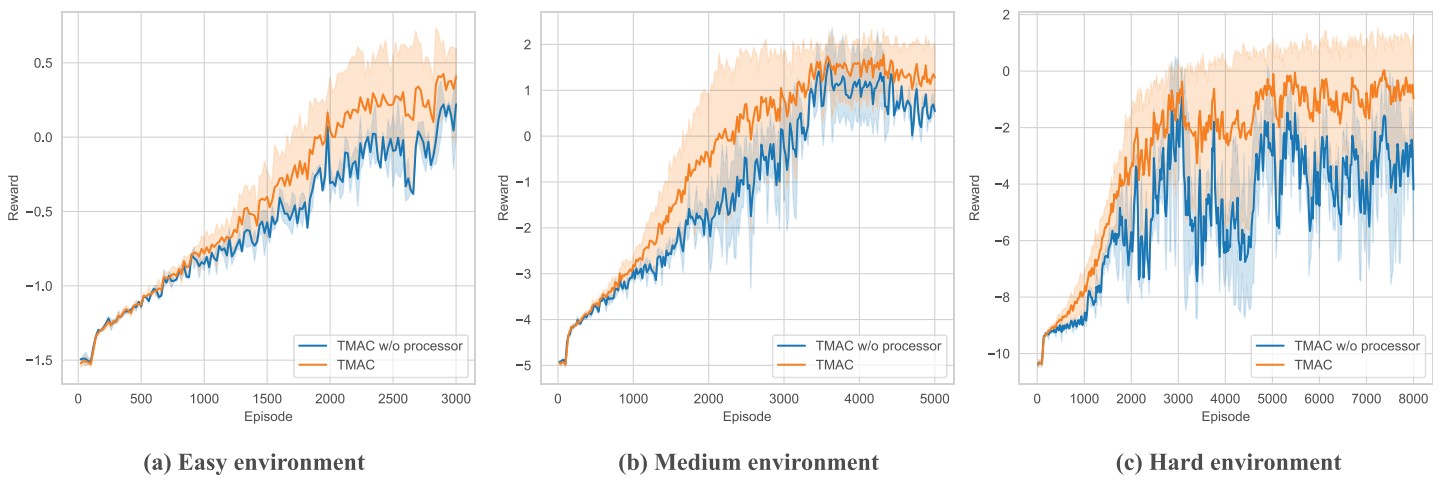

**(a) Easy environment**  **(b) Medium environment**  **(c) Hard environment**

**Figure 8** Performance comparison chart of our algorithm after removing the message processor module in easy, medium and hard environments.

**Table 3** Table of average rewards obtained by our algorithm and ablation experiments with partial module removal under three environment settings.

| Difficulty | TMAC w/o self-message fusing module | TMAC w/o message processor | TMAC w/MLP | TMAC |
|---|---|---|---|---|
| Easy | −0.33 ± 0.07 | −0.59 ± 0.01 | −0.49 ± 0.06 | **−0.25 ± 0.01** |
| Medium | −0.05 ± 0.05 | −1.13 ± 0.03 | −3.1 ± 0.17 | **0.22 ± 0.02** |
| Hard | −2.87 ± 0.26 | −5.01 ± 0.71 | −8.01 ± 0.3 | **−1.33 ± 0.3** |

**Note:**
Bold represents the best results.

performing ablation experiments on the message processor, we did not completely lose the message processing capability of the model but use a simple attention module to replace the original message processor.

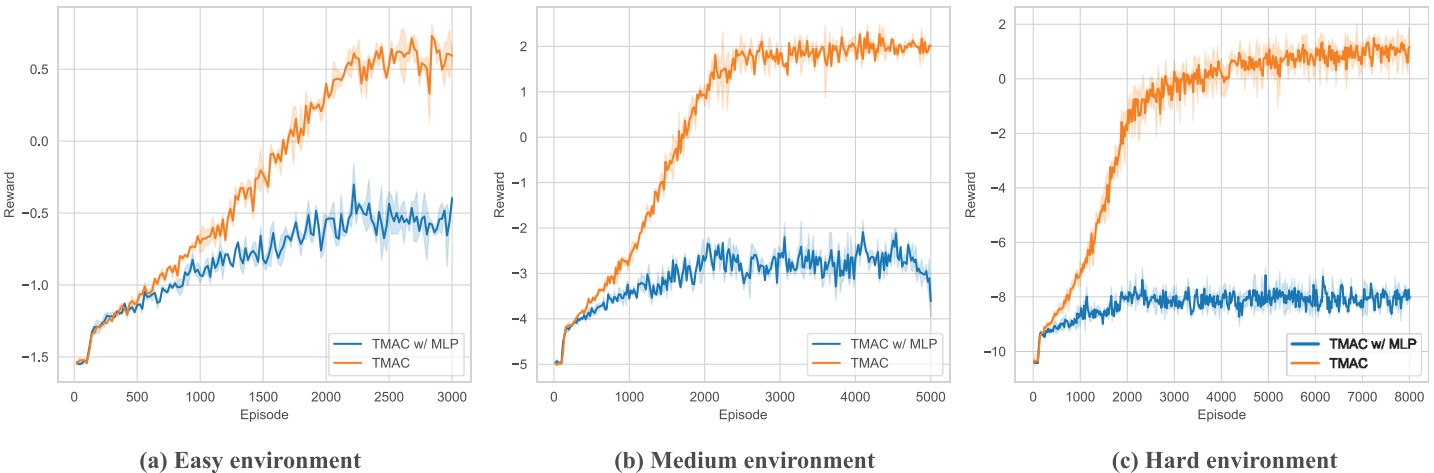

(a) Easy environment  (b) Medium environment  (c) Hard environment

**Figure 9 Performance comparison chart of our algorithm after replace feature extraction module with MLP in easy, medium and hard environments.**

As shown in Fig. 7 and Table 3, bold fonts represent better results in table, for the ablation experiment of the message self-encoding module, it is equivalent to the fact that the agent cannot efficiently extract useful messages from it every time it obtains an environmental observation, resulting in a decline in overall performance, and this phenomenon will increase with time. It becomes increasingly apparent as the difficulty of the environment increases. However, the agent does not entirely lose its ability to organize messages. This is because, in subsequent communications, the message processor will assist the agent in generating messages for communication.

As shown in Fig. 8 and Table 3, for the message processor ablation experiment, we replace the original message processor with a simple attention network. From the figure, we can find that the overall performance of the system has declined, and there is a more serious downward trend as the difficulty of the environment increases. This shows that our message processor module can significantly help the agent integrate and process messages from other agents, thereby assisting the agent to take correct actions. At the same time, experiments show that the message processor we proposed is better able to extract key messages from complex messages in complex environments, helping agents make better decisions.

As shown in Fig. 9 and Table 3, in order to verify the ability of TMAC to improve the agent's feature extraction, we replaced the feature extraction module (multi-head attention module) in TMAC with a multilayer perceptron (MLP) and tested it in surviving. The results showed that after losing the feature extraction module, the agent's performance dropped significantly.

## SUMMARY AND OUTLOOK

This article introduces TMAC, a multi-agent reinforcement learning communication protocol designed for locally observable environments. The message processor enables agents to perform multiple rounds of communication and integrate their existing messages at each round. Additionally, for multi-agent reinforcement learning in complex

environments, we propose a self-message fusion module that helps agents filter critical information by mapping observations from multiple dimensions to a unified one and extracting integrated features. Simulation experiments demonstrate that these structures significantly enhance communication efficiency in multi-agent reinforcement learning, resulting in higher reward feedback for the entire system.

Furthermore, the algorithm presented in this article primarily addresses the communication challenges in multi-agent collaborative scenarios. It recognizes the complexities introduced by environments where agents may compete or where both collaboration and competition coexist. This understanding lays the groundwork for future research, where we aim to explore these issues in greater depth, ultimately advancing more effective and efficient multi-agent collaboration.

In particular, when applying TMAC to real world tasks like a multi-drone coordination task, we woule face challenges in ensuring efficient communication between drones in dynamic environments, where network connectivity can fluctuate and communication delays may occur. Moreover, the computational load of each agent needs to be optimized to accommodate the limited hardware capabilities of drones, especially in large-scale operations.

### Funding
This work was supported by Major Research Project of National Natural Science Foundation of China under Grant 92267110, by NSFC under Grant 62476225, 62076202, and by National Key R&D Program of China under Grant 2023YFF0905604. There was no additional external funding received for this study. The funders had no role in study design, data collection and analysis, decision to publish, or preparation of the manuscript.

### Grant Disclosures
The following grant information was disclosed by the authors:
Major Research Project of National Natural Science Foundation of China: 92267110.
NSFC: 62476225, 62076202.
National Key R&D Program of China: 2023YFF0905604.

### Competing Interests
The authors declare that they have no competing interests.

### Author Contributions
- Xuesi Li conceived and designed the experiments, prepared figures and/or tables, and approved the final draft.
- Shuai Xue performed the experiments, performed the computation work, prepared figures and/or tables, and approved the final draft.
- Ziming He analyzed the data, authored or reviewed drafts of the article, and approved the final draft.

- Haobin Shi analyzed the data, authored or reviewed drafts of the article, and approved the final draft.

## Data Availability

The code is available at Zenodo: xue. (2025). tmac code. Zenodo. https://doi.org/10.5281/zenodo.14637380.

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
