# Peer review of "TMAC: a Transformer-based partially observable multi-agent communication method"

_PeerJ Computer Science, doi:10.7717/peerj-cs.2758_

## Round 0.1 · original submission · Major Revisions

Major revision is required for this paper based on lacking of some technical details in the presented work. Authors should also consider the following in the revised manuscript.

1. In "Abstract", authors should provide the quantitative performance achievement of the proposed method.
2. Carefully review the literature to cover the state-of-the-art works.
3. The details of why certain settings were selected should be clearly explained in their experiments.

Reviewer 1 ·

Basic reporting

This paper proposed a Transformer-based Multi-Agent Communication (TMAC) method for partially observable multiagent cooperation. The writing is good, and the figures and tables are clear. Moreover, the reviewer found that the self-message fusion algorithm is novel.

Points to improve:
1. some claims are not suported by the experiments, for example, the abstract writes that "TMAC improves agents extracting features and generating output messages", but experiments do not contains any extracting features analysis.
2. many claims are not properly citated with previous works. For example, the authors say that "Full communication may bring a large amount of redundant information", in fact, previous works have verified this. It is better to cite previous works.
3. Literature review is not sufficient. For example, early-era deep MA communication methods (e.g., ACCNet, attention-based MA communication methods, and MA communication with limited-bandwidth are not mentioned.
4. Experiment setting can be improved. For example, including Transformer-based Multi-Agent cooperation methods and testing on more environments

Experimental design

Overall, the experiments design are good, including main results and ablation study.

Points to improve:
1. including Transformer-based Multi-Agent cooperation methods since your method is a Transformer-based one.
2. testing on more environments (e.g., SMAC). Currently, only testing on a grid environment.
3. some key claims can be suported by the experiments, for example, the abstract writes that "TMAC improves agents extracting features and generating output messages", but experiments do not contains any extracting features analysis.

Validity of the findings

no comment

Cite this review as

Reviewer 2 ·

Basic reporting

The paper introduces TMAC, a Transformer-based communication algorithm for multi-agent systems under partial observability. It aims to enhance feature extraction and message generation in communication-restricted environments. It proposes a self-message fusion module to integrate information from various sources, thereby facilitating more effective coordination among agents. Please see below my comments:

Experimental design

5. Authors doesn’t explain why certain settings, like the number of Transformer layers, were chosen.
6. More thought on how TMAC manages message passing would be helpful, especially if many agents are involved at once, as indicated by my previous comment.

Validity of the findings

3. Transformers are known to use a lot of computing power, especially with many agents involved. It would be better if the paper indicates whether TMAC’s approach can work in settings that don’t have as many resources or need to run on simpler devices.
4. Since the paper only uses a single grid simulation for testing, it’s hard to tell if TMAC will work in other setups. Please clarify.

Additional comments

1. Since the paper addresses an important topic related to effective communication and sensing, the paper should be related to upcoming 6G generation of wireless. In particular, in multi-agent systems where agents need to sense their surroundings and communicate with each other simultaneously, 6G's ISAC [REF01] technology allows for seamless integration of both functions over shared infrastructure. This is especially relevant for partially observable environments in MARL, where sensing is as crucial as communication for agents to build an understanding of the environment, share situational awareness, and coordinate actions.
2. The paper can discuss more about the practical difficulties of using TMAC with real-world systems.
7. The conclusion may be enhanced.


References
[REF01] A. Bazzi and M. Chafii, “On Outage-Based Beamforming Design for Dual-Functional Radar-Communication 6G Systems,” in IEEE Transactions on Wireless Communications, vol. 22, no. 8, pp. 5598-5612, Aug. 2023, doi: 10.1109/TWC.2023.3235617

Cite this review as

---

## Round 0.2 · accepted · Accept

Authors have addressed all the comments from the reviewers, Hence, it is recommended to accept this paper in its current form.

Reviewer 1 ·

Basic reporting

The new version addressed most of my concerns.

Experimental design

The new version addressed most of my concerns.

Validity of the findings

The new version addressed most of my concerns.

Cite this review as

Reviewer 2 ·

Basic reporting

no comment
The authors have addressed my comments and i suggest accepting the paper

Experimental design

no comment
The authors have addressed my comments and i suggest accepting the paper

Validity of the findings

no comment
The authors have addressed my comments and i suggest accepting the paper

Additional comments

no comment
The authors have addressed my comments and i suggest accepting the paper

Cite this review as